# Blending Two Styles: Generating Inter-domain Images with MiddleGAN

**Collin MacDonald**[*]                                      *cmacdonald01@wm.edu*
*Department of Computer Science*
*William & Mary*

**Zhendong Chu**                                              *zc9uy@virginia.edu*
*Department of Computer Science*
*University of Virginia*

**John Stankovic**                                           *jas9f@virginia.edu*
*Department of Computer Science*
*University of Virginia*

**Huajie Shao**                                                *hshao@wm.edu*
*Department of Computer Science*
*William & Mary*

**Gang Zhou**                                                  *gzhou@wm.edu*
*Department of Computer Science*
*William & Mary*

**Ye Gao**[*][†]                                              *ygao18@wm.edu*
*Department of Computer Science*
*William & Mary*

**Reviewed on OpenReview:** *https://openreview.net/forum?id=t7vWCHmwbG*

## Abstract

From celebrity faces to cats and dogs, humans enjoy pushing the boundaries of art by blending existing concepts together in new ways. With the rise of generative artificial intelligence, machines are increasingly capable of creating new images. Generative Adversarial Networks (GANs) generate images similar to their training data but struggle to blend images from distinct datasets. This paper introduces MiddleGAN, a novel GAN variant that blends inter-domain images from two distinct input sets. By incorporating a second discriminator, MiddleGAN forces the generator to create images that fool both discriminators, thus capturing the qualities of both input sets. We also introduce a blend ratio hyperparameter to control the weighting of the input sets and compensate for datasets of different complexities. Evaluating MiddleGAN on the CelebA dataset, we demonstrate that it successfully generates images that lie between the distributions of the input sets, both mathematically and visually. An additional experiment verifies the viability of MiddleGAN on handwritten digit datasets (DIDA and MNIST). We provide a proof of optimal convergence for the neural networks in our architecture and show that MiddleGAN functions across various resolutions and blend ratios. We conclude with potential future research directions for MiddleGAN.

---

[*]Equal contribution.
[†]Corresponding author.

# 1 Introduction

Humans have always loved to create, especially visually. Using advanced photo manipulation tools such as Photoshop, digital artists have produced highly realistic blended faces of well-known celebrities (Emma Taggart). Simultaneously, with the rise of generative artificial intelligence (AI), computers are now "creating" images and other artistic works as well (Cetinic & She). One such method of image creation is Generative Adversarial Networks (GANs). GANs are machine learning models that are trained on a set of images and learn to create new images (from random noise) that are similar to the training images. More formally, GANs learn to generate images that are in distribution with the images of the input domain. The use of GANs to create images has been extensively explored over the last decade. However, traditional GANs are not designed to blend two distinct sets of images together.

In this paper, we present MiddleGAN, a novel variation of the traditional GAN, which takes as input two domains of images and learns to create images that lie between the input domains. In essence, MiddleGAN aims to create blended inter-domain images that contain features from both input domains. For example, given the input domains of male and female faces, MiddleGAN can produce images of human faces with both masculine and feminine qualities, as shown in Figure 1. Compared to existing state-of-the-art multi-discriminator GAN variations, such as FairGAN (Xu et al.) and D2GAN (Nguyen et al.), the dual-dataset input and blended-image objective of MiddleGAN is novel. Instead of manually blending the faces of female and male celebrities, an artist could use a model such as MiddleGAN to create entirely new faces at the click of a button. Artists and designers can use the blended features to create unique and original artworks. By generating faces or figures that combine male and female traits, they can explore new aesthetic possibilities or challenge traditional gender representations in art and media. In addition, in industries like gaming and animation, blending features from different domains (e.g., male and female) can allow for more nuanced character creation. It could help generate characters with androgynous features or those that evolve across different gender traits dynamically.

To achieve our goal of inter-domain image generation, we ask three overarching research questions:

- **RQ1:** How can we create inter-domain images that have qualities of both input domains, and validate this methodologically via t-SNE?

- **RQ2:** Is it feasible to generate images that lie in the middle of the feature space of the two domains, or can we extend our methodology to place unequal emphasis on each input domain?

- **RQ3:** Are the generated images consistently high quality across multiple image sizes and blend ratios?

In order to answer these research questions, we make three contributions:

- First, we propose a novel variation of a GAN, MiddleGAN, which leverages two discriminators and one generator to create blended inter-domain images from two input datasets.

- Second, we extend the initial functionality of MiddleGAN to introduce a Blend Ratio, which controls the amount of emphasis placed on each input domain's corresponding discriminator.

- Third, through extensive evaluation on the CelebA dataset, we show that images generated by MiddleGAN are within the expected distribution (via t-SNE) and have both masculine and feminine qualities when visually examined.

The remainder of this paper is organized as follows: Section 2 describes related work and the context of this project. Section 4 describes the theoretical basis of MiddleGAN and its model architecture. Section 5 discusses our implementation of MiddleGAN. Section 6 details our extensive evaluation of MiddleGAN. Section 7 discusses MiddleGAN and outlines opportunities for future work. Section 8 concludes our paper.

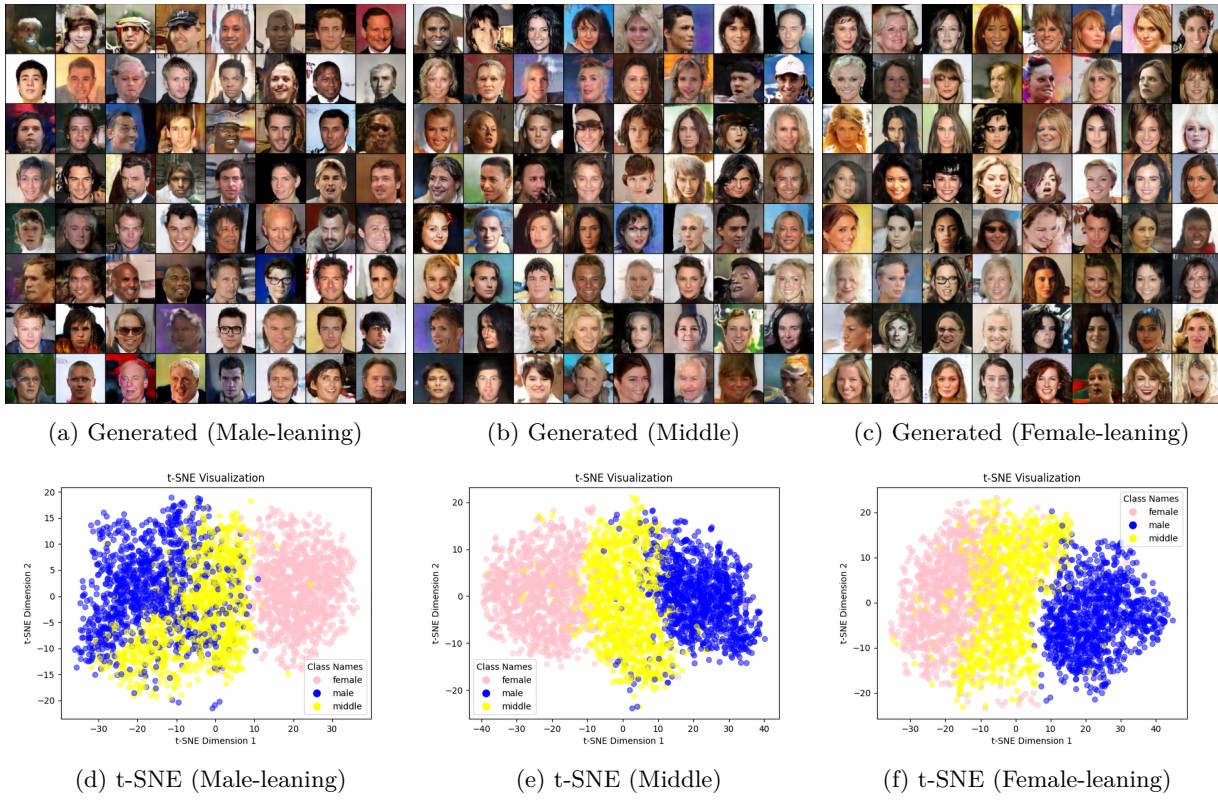

(a) Generated (Male-leaning)   (b) Generated (Middle)   (c) Generated (Female-leaning)

(d) t-SNE (Male-leaning)   (e) t-SNE (Middle)   (f) t-SNE (Female-leaning)

Figure 1: MiddleGAN can generate images which are blends of two domains. In this figure, we show the results of images generated with MiddleGAN at a resolution of 128px. The t-SNE visualizations support our claim that the blended images fall within the two input distributions (male and female).

## 2 Related Work

In this section, we present an overview of the original generative adversarial network (GAN) as well as several subsequent GANs as they relate to our current work.

**Generative Adversarial Nets:** Since the introduction of the first generative adversarial network over a decade ago, many novel variations of GANs have been introduced. However, at the start of them all is the work of Goodfellow et al. in the paper "Generative Adversarial Nets" (Goodfellow et al.). The original GAN contained one discriminator and one generator. These models are trained in parallel. The generator learns to take random noise as input and produce an image as output. The discriminator learns to distinguish between real images and those created by the generator. By engaging in a minimax game, the generator learns to produce better images while the discriminator learns to better differentiate between real and generated images. The goal of training is to create a generator model that can produce images, from uniform random noise, that are within the distribution of the original training images. In other words, "GANs are a framework for teaching a deep learning model to capture the training data distribution so we can generate new data from that same distribution" (Tutorials).

GANs have been used in a wide range of applications, ranging from image-to-image style transfer to domain transfer tasks. CycleGAN focuses on "unpaired image-to-image translation," while GP-GAN focuses on high-resolution composite image blending (Wu et al.; Zhu et al.). The work of Sankar et al. focuses on the use of GANs for domain adaptation, and the work of Rahman et al. focuses on how GANs can be leveraged for domain generalization.

**DCGAN:** While an important first step, the original GAN as proposed by Goodfellow et al. is notoriously hard to train and can often fail to converge (Nie & Patel). Due to these challenges, many improved GAN

variations have been proposed over the course of the last decade. One such proposed improvement is the Deep Convolutional Generative Adversarial Network (DCGAN) (Radford et al.). Radford et al. modified the original GAN to use Convolutional Neural Networks (CNN), which at the time were gaining popularity in computer vision tasks. DCGAN forms the basis for our MiddleGAN model, with its use of "convolutional and convolutional-transpose layers in the discriminator and generator, respectively" (Tutorials). DCGAN helped to prove the viability of CNN-based GANs and improved the training stability of the model when compared to the original GAN (Radford et al.).

**WGAN & WGAN-GP:** The Wasserstein GAN (WGAN), introduced by Arjovsky et al. in 2017, builds upon the success of DCGAN, making use of a CNN-based architecture (Arjovsky et al.). The novelty of the WGAN is two-fold. First, WGAN trains the discriminator (termed "critic") at a higher rate than the generator. As such, "training WGANs does not require maintaining a careful balance in training of the discriminator and the generator, and does not require a careful design of the network architecture either" (Arjovsky et al.). Second, and more importantly, the loss function for the WGAN is based on the Earth Mover's (Wasserstein-1) distance, which differs from traditional GANs and provides "a meaningful loss metric that correlates with the generator's convergence and sample quality" (Arjovsky et al.).

Gulrajani et al. proposed WGAN-GP as a direct improvement of the traditional Wasserstein GAN (Gulrajani et al.). Instead of using weight clipping to enforce the Lipschitz constraint on the discriminator/critic, which is required by WGAN, WGAN-GP "penalizes the norm of the gradient of the critic with respect to its input" (Gulrajani et al.). We use the loss function of WGAN-GP as the basis for the loss function of MiddleGAN.

**Multi-Discriminator GANs:** Beyond the above works, several other proposed GAN variations make use of dual discriminators. In "Generative Multi-Adversarial Networks," Durugkar et al. propose the use of multiple discriminators for a variety of use cases, including serving as a "formidable adversary" and a "forgiving teacher" (Durugkar et al.). Nguyen et al. attempt to mitigate issues of mode collapse through the use of dual discriminators in "Dual Discriminator Generative Adversarial Nets" (Nguyen et al.). More concretely, Xu et al. make use of two discriminators to improve the fairness of generated images with FairGAN (Xu et al.). MD-GAN proposes the use of multiple discriminators as an approach to distributed learning (Hardy et al.). Lastly, PATE-GAN uses multiple discriminators as applied to the field of differential privacy (Jordon & Yoon).

**Diffusion Models:** Given the introduction of GANs in 2014, they are somewhat dated. One might be tempted to ask why we focused on a GAN-based architecture as opposed to a more modern architecture, such as Diffusion Models. Diffusion Models, introduced in 2015, leverage ideas from "nonequilibrium thermodynamics" to train a generative AI model (Sohl-Dickstein et al.). Unlike GANs, diffusion models are not adversarial. Instead, diffusion models work by "systematically and slowly destroying structure in a data distribution through an iterative forward diffusion process... [and] then learning a reverse diffusion process that restores structure in data, yielding a highly flexible and tractable generative model of the data" (Sohl-Dickstein et al.). In other words, a diffusion model first learns to convert an image to seemingly random noise, and then learns to reverse this process, taking random noise and converting it into an in-distribution image. This process cannot be easily extended to support multiple input datasets, which prevents traditional diffusion models from forming the basis for an inter-domain image generation model. While more recent work such as "Blended Latent Diffusion" has leveraged diffusion models to mask and blend individual images, this work does not overcome the inherent single-domain limitation of diffusion models (Avrahami et al.).

When compared to previous works, MiddleGAN is distinct in its goal. Unlike traditional GANs, MiddleGAN utilizes two discriminators and one generator. Even when compared to other multi-discriminator GANs such as MD-GAN and FairGAN, the objective of MiddleGAN is noticeably different, as MiddleGAN aims to generate images that are in between the distributions of the two input datasets. Despite the existence of more modern generative AI architectures such as diffusion models, MiddleGAN is still relevant as it leverages the adversarial nature of GANs to generate images in between two distributions, which diffusion models cannot do.

## 3 Theory

In Section 4, we are going to provide detailed information on exactly how we achieve to obtain the inter-domain. In this Section (Section 3), we explore what is the potential usage of the inter-domain. To wit, the inter-domain, whose similarity to the two original domains is showcased by being equally distant (measured via Wasserstein distance) to the two original domains, can be used as an intermediate domain shown to facilitate better domain adaptation and generalization: existing works Na et al. (2021); Wang et al. (2022) have already showcased that empirically, adapting from one original domain (source) to another (target) is useful. Here, we theoretically prove that the inter-domain generated by MiddleGAN can indeed lower the error difference over distribution and domain shift if we adapt from one original domain to the inter-domain then to the other original domain, compared with the strategy in which the domain adaptation/generalization happens directly from one original domain (that is used as the source) to the other original domain (that is used as the target). The purpose of this proof is to showcase a potential usage for the generated synthetic inter-domain.

### 3.1 Preliminaries

Let $\mathcal{X}$ and $\mathcal{Y}$ represent the input and output spaces, respectively. Let $X$ and $Y$ be random variables taken from the input and output spaces. For a given domain (e.g., female faces), the distribution is denoted as $\mu$ over $\mathcal{X} \times \mathcal{Y}$. When considering only the sample distribution and not the joint sample-label distribution, we denote the sample distribution of $\mu$ over the input space $\mathcal{X}$ as $\mu(X)$.

**Assumption 1 (Bounded Input Space)**. A compact input space $\mathcal{X}$ is bounded in the d-dimensional unit $\ell_2$ ball:

$$\mathcal{X} \subseteq \{x \in \mathbb{R}^d : \|x\|_2 \leq 1\}$$

**Definition 1 (p-Wasserstein Distance)**. Consider $\mu$ and $\nu$ over $\mathcal{S} \subset \mathbb{R}^d$. For any $p \geq 1$, the $p$-Wasserstein distance is defined as:

$$W_p(\mu, \nu) := \left( \inf_{\gamma \in \Gamma(\mu, \nu)} \int_{\mathcal{S} \times \mathcal{S}} d(x, y)^p \, d\gamma(x, y) \right)^{1/p}$$

where $\Gamma(\mu, \nu)$ is the set of all measures over $\mathcal{S} \times \mathcal{S}$ with marginals equal to $\mu$ and $\nu$ respectively.

### 3.2 Error Difference over Distribution & Domain Shift

**Assumption 2 ($R$-Lipschitz Classifier)**. Let $h \in \mathcal{H}$ be $R$-Lipschitz in $\ell_2$ norm, i.e.:

$$\forall x, x \in \mathcal{X} : |h(x) - h(x')| \leq R\|x - x'\|_2$$

**Assumption 3 ($\rho$-Lipschitz Loss)**. $\ell$ is $\rho$-Lipschitz, i.e., $\forall y, y' \in \mathcal{Y}$:

$$|\ell(y, \cdot) - \ell(y', \cdot)| \leq \rho\|y - y'\|_2$$

$$|\ell(\cdot, y) - y(\cdot, y)| \leq \rho\|y - y'\|_2$$

In the settings of MiddleGAN, consider $\mu, \nu$ and the measures for the middle domain $m$.

**Lemma 1 (Error difference over shifted domains):**

Population loss $|\epsilon_\mu(h) - \epsilon_m(h)| + |\epsilon_m(h) - \epsilon_\nu(h)| \leq \rho\sqrt{R^2 + 1}W_p(\mu, \nu)$

**Proof:**

$$|\epsilon_\mu(h) - \epsilon_m(h)| = |\mathbb{E}_{x,y\sim\mu}[\ell(h(x),y)] - \mathbb{E}_{x',y'\sim m}[\ell(h(x'),y')]|$$

$$= \left|\int \ell(h(x),y)\,d\mu - \int \ell(h(x'),y')\,dm\right|$$

$$|\epsilon_m(h) - \epsilon_\nu(h)| = |\mathbb{E}_{x',y'\sim m}[\ell(h(x'),y')] - \mathbb{E}_{x'',y''\sim\nu}[\ell(h(x''),y'')]|$$

$$= \left|\int \ell(h(x'),y')\,dm - \int \ell(h(x''),y'')\,d\nu\right|$$

$$|\epsilon_\mu(h) - \epsilon_m(h)| + |\epsilon_m(h) - \epsilon_\nu(h)| = \left|\int \ell(h(x),y)\,d\mu - \int \ell(h(x'),y')\,dm\right| + \left|\int \ell(h(x'),y')\,dm - \int \ell(h(x''),y'')\,d\nu\right|$$

Let $\gamma$ be an arbitrary coupling of $\mu$ and $m$ (i.e., a joint distribution of $\mu, m$). Similarly, we define $\gamma_{m,\nu}$.

$$|\epsilon_\mu\ell(h) - \epsilon_\nu\ell(h)| = \left|\int \ell(h(x),y) - \ell(h(x'),y')\,d\gamma_{\mu,m}\right|$$

Triangle inequality $\leq \int \left|\ell(h(x),y) - \int \ell(h(x'),y')\right|\,d\gamma_{\mu,m}$

($\ell$ is $\rho$-Lipschitz)

$$\left|\int \ell(h(x),y) - \ell(h(x'),y')\,d\gamma_{\mu,m}\right| \leq \int \rho(||h(x) - h(x')||) + \rho||y - y'||)\,d\gamma_{\mu,m}$$

($h$ is $R$-Lipschitz)

$$\int \rho(||h(x) - h(x')||) + \rho||y - y'||)\,d\gamma_{\mu,m} \leq \int \rho R||x - x'|| + \rho||y - y'||\,d\gamma_{\mu,m}$$

(R > 0)

$$\leq \int \rho\sqrt{R^2 + 1}(||x - x'|| + ||y - y'||)\,d\gamma_{\mu,m}$$

$$\leq \inf_{\gamma_{\mu,m}} \int \rho\sqrt{R^2 + 1}(||x - x'|| + ||y - y'||)\,d\gamma_{\mu,m}$$

$$= \rho\sqrt{R^2 + 1}W_1(\mu,m) \leq \rho\sqrt{R^2 + 1}W_p(\mu,m)$$

Similarly:

$$|\epsilon_m(h) - \epsilon_\nu(h)| \leq \rho\sqrt{R^2 + 1}W_p(m,\nu)$$

$$|\epsilon_\mu(h) - \epsilon_m(h)| + |\epsilon_m(h) - \epsilon_\nu(h)| \leq \rho\sqrt{R^2 + 1}W_p(\mu,m) + \rho\sqrt{R^2 + 1}W_p(m,\nu)$$

Since $W_p(\mu,m) = W_p(m,\nu) = \frac{1}{2}W_p(\mu,\nu)$:

$$|\epsilon_\mu(h) - \epsilon_m(h)| + |\epsilon_m(h) - \epsilon_\nu(h)| \leq 2\rho\sqrt{R^2 + 1}W_p(\mu,m) \leq \rho\sqrt{R^2 + 1}W_p(\mu,\nu)$$

**Remark:** Existing work Wang et al. (2022) already proves that $|\epsilon_\mu(h) - \epsilon_\nu(h)| \leq \rho\sqrt{R^2 + 1}W_p(\mu, \nu)$. Here, we prove that if there's a middle domain, $m$, then $|\epsilon_\mu(h) - \epsilon_m(h)|$ and $|\epsilon_m(h) - \epsilon_\nu(h)|$ are both $\leq \frac{1}{2}\rho\sqrt{R^2 + 1}W_p(\mu, \nu)$. Therefore, adapting from $\mu$ to $m$ and $m$ to $\nu$ breaks the original adaptation task from $\mu$ to $\nu$ in two subtasks, with each subtask equipped with smaller population loss on error difference over shifted domains.

## 4 Model Architecture & Design

In this section, we provide an overview of the model architecture as well as the design of MiddleGAN. Additionally, we provide a theoretical basis for MiddleGAN, building on top of the original GAN. We show show that there exists optimal parameters for both discriminators as well as an optimal solution for the parameters of the generator.

Before we discuss MiddleGAN, we need to discuss the original GAN on which MiddleGAN is based. In the original GAN (Goodfellow et al.), the generator $G$ and the discriminator $D$ engage in a minimax game in which $G$ tries to minimize a value objective $V(G, D)$ whereas $D$ tries to maximize it. $V(G, D)$ is defined in Equation 1, in which $p$ is the distribution of the real samples and $q$ is the distribution of the noise. A key observation obtained from Equation 1 is that $G$'s effort is to generate $G(z)$ whereas $z$ is an input noise such that $G(z)$ will be in-distribution with the distribution of the real samples $p$.

$$\min_G \max_D V(G, D) = \mathbb{E}_{x \sim p(x)}[\log(D(x)) \\ + \mathbb{E}_{z \sim q(z)}[\log(1 - D(G(z)))] \tag{1}$$

Based on the key observation that we obtain from Equation 1, in MiddleGAN we propose to employ two discriminators, $D_a$ and $D_b$. Each discriminator is only aware of one domain of images. The first discriminator, termed $D_a$, only knows of the images in Domain A. Similarly, the second discriminator, termed $D_b$, only knows of the images in Domain B. Neither discriminator knows of the existence of the other domain of images, and is solely responsible for determining if an image came from its corresponding domain or not. The generator $G$ engages in a two-way minimax game with the two discriminators. The samples it generates will be in the middle of the feature space of both input domains. Below, we empirically prove that the generated samples in $p_m$ are represented by the features that are invariant across the input domains.

Formally, the objective function of $D_a$, $D_b$, and $G$ is described by Equation 2.

$$\min_G \max_{D_a, D_b} V(G, D_a, D_b) \\ = \mathbb{E}_{x_a \sim p_a(x_a)}[\log(D(x_a))] + \mathbb{E}_{z \sim q(z)}[\log(1 - D_a(G(z)))] \\ + \mathbb{E}_{x_b \sim p_b(x_b)}[\log(D(x_b))] + \mathbb{E}_{z \sim q(z)}[\log(1 - D_b(G(z)))] \tag{2}$$

In the previous paragraphs we have described how to generate samples that are similar to samples from both input domains. Figure 2 outlines the duel-discriminator architecture of MiddleGAN and how it is used to generate domain agnostic samples.

### 4.1 Optima for the Discriminators and Generator

We first discuss the two discriminators $D_a$ and $D_b$ given a fixed $G$. We propose Theorem 4.1 regarding the optimal values for $D_a$ and $D_b$, represented as $D_a^*$ and $D_b^*$

Given $p_m$, the distribution of samples generated by a fixed generator $G$, the optimal values for the parameters of $D_a$ and $D_b$ are $D_a^* = \frac{p_a}{p_a + p_m}$ and $D_b^* = \frac{p_b}{p_b + p_m}$.

*Proof.* The value objective $V(G, D_a, D_b)$ can be expanded.

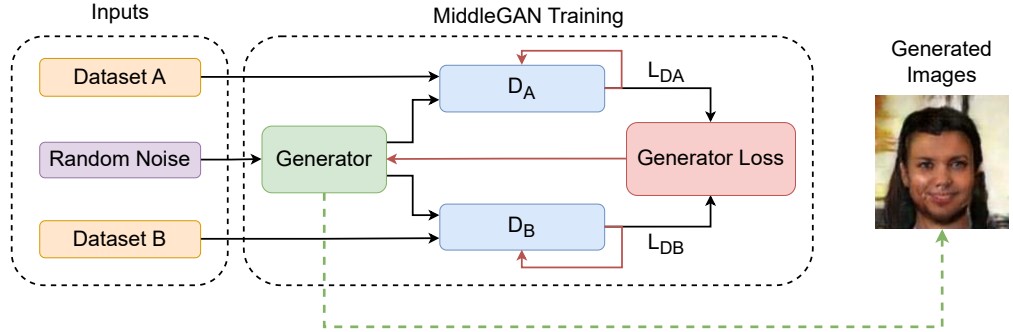

Figure 2: In this figure we describe how to use the MiddleGAN to generate fake, domain agnostic samples. As shown in the diagram, neither discriminator is directly aware of the others existence or the existence of a second dataset. This is importance, as it allows each discriminator to focus solely on differentiating between images in it's own dataset and any other images, regardless of source. Black arrows indicate the forward flow of information, while red arrows indicate backwards propagation. The dashed green line indicates that the generated images come from the trained generator.

$$
\begin{aligned}
V(G, D_a, D_b) &= \int_{x_a} p_a(x_a) \log(D_a(x_a)) dx_a + \int_{x_b} p_b(x_b) \log(D_b(x_b)) dx_b \\
&+ \int_z q(z) \log(1 - D_a(G(z))) dz + \int_z q(z) \log(1 - D_b(G(z))) dz \\
&= \int_{x_a} p_a(x_a) \log(D_a(x_a)) dx_a + p_m(x_a) \log(1 - D_a(x_a)) dx_a \\
&+ \int_{x_b} p_b(x_b) \log(D_b(x_b)) + p_m(x_b) \log(1 - D_a(x_b)) dx_b
\end{aligned}
\tag{3}
$$

We observe that $p_a$, $p_b$ and $p_m$ belong in $\mathbb{R}$. For the domain A discriminator $D_a$, any pair of $p_a$ and $p_m$ in the form of $p_a \log(y) + p_m (1 - \log(y))$, $p_a \log(y) + p_m (1 - \log(y))$ achieves its maximum value at $\frac{p_a}{p_a + p_m}$ (Goodfellow et al.). Similarly, for the domain B discriminator $D_b$, any pair of $p_b$ and $p_m$ in the form of $p_b \log(y) + p_m (1 - \log(y))$, $p_b \log(y) + p_m (1 - \log(y))$ achieves its maximum value at $\frac{p_b}{p_b + p_m}$. $\qquad \square$

Now we bring forth Theorem 4.1 which proposes that there exists an optimal solution for the parameters of not only $D_a$ and $D_b$, but also $G$. There exists a global minimum for the virtual training criterion $C(G)$ defined as

$$
C(G) = \max_{D_a, D_b} V(G, D_a, D_b).
\tag{4}
$$

In other words, there exists an optimal solution for the parameters of the generator $G$.

*Proof.* Goodfellow et al. proved that, in the original GAN where there is only one discriminator $D$ and one generator $G$, the virtual training criterion can be written as the following:

$$
\begin{aligned}
C_{original}(G) &= \max_D V(G, D) \\
&= -log(4) + KL(p \parallel \frac{p + p_m}{2}) + KL(p_m \parallel \frac{p + p_m}{2})
\end{aligned}
\tag{5}
$$

in which $p$ is the distribution of the real samples and $p_m$ is the distribution of generated fake samples, and KL is the Kullback–Leibler divergence. With two discriminators, our virtual training criterion $C(G)$ can be rewritten as:

$$C(G) = -log(4) + KL(p_a \parallel \frac{p_a + p_m}{2}) + KL(p_m \parallel \frac{p_a + p_m}{2})$$
$$- log(4) + KL(p_b \parallel \frac{p_b + p_m}{2}) + KL(p_m \parallel \frac{p_b + p_m}{2}) \tag{6}$$
$$= -2log(4) + 2JSD(p_a \parallel p_m) + 2JSD(p_b \parallel p_m)$$

In Equation 6, JSD is the Jensen–Shannon divergence. To find the global minimum, $M(G)$, we want to obtain

$$M(G) = \underset{p_m}{arg\,min} - 2log(4) + 2JSD(p_a \parallel p_m) + 2JSD(p_b \parallel p_m) \tag{7}$$

We observe in Equation 7 that we are looking for the optimal value of the JSD centroid defined as $Centroid^* = arg\,\underset{Q}{min} \sum_{i=1}^{n} JSD(P_i \parallel Q)$ in which $P_i$ and $Q$ are distributions. We can see that the generator is essentially looking for the JSD controid of the source domain distribution $p_a$ and the target domain distribution $p_b$. The convexity of the problem has been proved in (Nielsen, 2020). □

## 4.2 Beyond the "Middle" Feature Space

With two discriminators, the generator $G$ faces an additional challenge, as its success is determined by its ability to fool both discriminators. In order to do this, the generator must learn to produce images that have features of both input domains. Simply put, the loss of the generator is the averaged loss from both discriminators, as shown in Equation 8.

$$\mathcal{L}_G = \frac{(\mathcal{L}_{D_A} + \mathcal{L}_{D_B})}{2} \tag{8}$$

In Equation 8, the losses of both discriminators are weighed equally. This aims to ensure that the generator $G$ learns to produce images that are in the middle of both input domains. However, this is an artificial limitation, as the weights for the losses from each discriminator do not have to be evenly weighted. To expand the flexibility of MiddleGAN, we introduced a new blending hyperparameter, the blend ratio, into our model architecture.

$$\mathcal{L}_G = \mathcal{L}_{D_A} * (BR) + \mathcal{L}_{D_B} * (1 - BR) \tag{9}$$

The updated loss for the generator is shown in Equation 9. The blend ratio can range from 0 to 1, and directly controls the percent of weight placed on the loss of $D_A$. A blend ratio of 0.5 equally weighs the losses from both discriminators and replicates our initial implementation of MiddleGAN. A blend ratio of 0 would completely ignore Domain A and cause MiddleGAN to act like a traditional GAN, using only Domain B as input. A blend ratio of 0.75 would result in the loss of discriminator A carrying three times the weight of the loss of discriminator B. The blend ratio vastly increases the range of images that MiddleGAN can learn to produce.

### 4.2.1 Choosing the Blending Ratio.

In many generative models, the influence of domains is affected not only by the blend ratio but also by factors such as the underlying data distributions and the number of samples. However, in MiddleGAN, the blend ratio directly determines how the losses from the two discriminators are weighted, enabling us to systematically blend features from both domains without relying on assumptions about the data distributions. Since the discriminators are trained independently on their respective datasets, the blend ratio effectively serves as a weighted "vote" between the loss functions, providing more precise control over the blending process than the raw characteristics of the data alone.

We agree that manually selecting the appropriate blend ratio can be inefficient. To address this, we suggest using grid search to systematically optimize the blend ratio. Our aim is to minimize metrics like Kernel

Inception Distance (KID) and Fréchet Inception Distance (FID), making grid search a logical approach. By exploring different blend ratio values, we can objectively determine the best ratio that balances the contributions of both domains and enhances image quality.

However, an important point to note is that while FID and KID are typically used to assess the similarity between real and generated images, MiddleGAN's goal is different. Instead of producing images that perfectly match a single real dataset, we aim to generate images that intentionally blend features from both domains. This means that our generated images are intentionally distinct from either of the real datasets. For details on the FID and KID scores after grid search, please refer to Table 2 in Section 6.2.1.

### 4.3 Novelty of MiddleGAN

At first glance, MiddleGAN might appear similar to a traditional GAN, but with two discriminators instead of one. However, the second discriminator plays a pivotal role that sets MiddleGAN apart from standard GANs. While many GAN variants utilize multiple discriminators, MiddleGAN is uniquely designed to generate inter-domain images that blend features from two distinct datasets, which is not the primary focus of typical GANs. The second discriminator is not merely an incremental addition; it is crucial for ensuring that the generated images effectively combine characteristics from both input domains. This enables MiddleGAN to function in a space between domain translation and image synthesis, providing a solution for generating images that exist between two feature distributions—something that traditional GANs, like DCGAN or StarGAN, are not built to do. Additionally, the blend ratio hyperparameter enhances the model's flexibility, allowing precise control over the influence of each domain. This feature makes MiddleGAN particularly valuable in scenarios where a balance between characteristics from two domains is required, representing a significant advancement over conventional GAN approaches.

## 5 Implementation

Our implementation of MiddleGAN builds upon PyTorch's official implementation of DCGAN (Tutorials). This implementation was designed for use with the CelebA dataset, which sped up our initial development time. However, this implementation had several limitations, most notably a fixed input image size of 64px. By dynamically computing the number of layers required by both the generator and discriminator models based on the requested image size, we enabled our implementation of DCGAN to support any power-of-two image size.

Next, we modified our implementation of DCGAN to add an additional discriminator and support two input domains. These changes formed the basis for MiddleGAN. We modified the model architecture so that in each training epoch, images from both domain A and domain B were seen by the model. To account for differences in the sizes of the input domains, we leveraged the itertools library's Cycle method to ensure that the length of the smaller dataset would appear to match that of the larger dataset (pyt, a). With the addition of the second discriminator, we simply adjusted the loss function for the generator to take into account the losses of both discriminators when evaluated on generated images. At this stage, our implementation of MiddleGAN could generate images, albeit with low quality.

With the basic MiddleGAN implementation complete, we began preliminary testing and evaluation of the model. We observed high training instability of the GAN and poor quality of generated images. After making these observations, we began to look into ways to stabilize the training of the GAN.

In parallel, we were exploring the idea of a transformer-based implementation of MiddleGAN, based on TransGAN (Jiang et al.). While the paper authors provided a PyTorch implementation of TransGAN (git, b) (which itself was based on their earlier work, AutoGAN (git, a)), we opted for a cleaner re-implementation of TransGAN (Sarıgün). Using TransGAN as our base model, we developed a low resolution (32px) transformer-based version of MiddleGAN. This implementation utilized the loss function from WGAN-GP (Gulrajani et al.) to stabilize the training process. A sample of the images produced with our transformer-based implementation of MiddleGAN is shown in Figure 3a. Figure 3c and Figure 3b were very intermixed, showing that generated samples were not clearly between the input distributions. Given these results, along with significantly longer training times we observed, we ultimately declined to pursue development

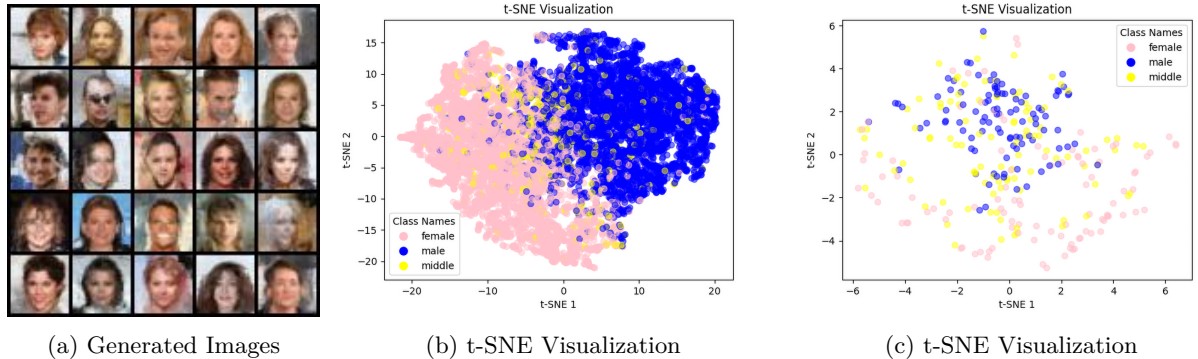

(a) Generated Images      (b) t-SNE Visualization      (c) t-SNE Visualization

Figure 3: A sample of images generated with our transformer-based MiddleGAN implementation as well as associated tSNE visualizations. The images were generated at a resolution of 32px by 32px with a blend ratio of of 0.50. **As shown by the t-SNE visualizations, the generated images fell within the distribution of both input domains (as opposed to between it).**

of a transformer-based MiddleGAN for this project. We leave it to future research to better evaluate the viability and effectiveness of a transformer-based MiddleGAN.

While we ultimately declined to pursue a Transformer-based implementation of MiddleGAN, that work was not in vain, as we did observe that the loss function being used in TransGAN was leading to more stable model training. The loss function, WGAN-GP loss, is based on the work of Gulrajani et al. in "Improved Training of Wasserstein GANs" (Gulrajani et al.). When we implemented WGAN-GP loss on our DCGAN-based MiddleGAN, we observed more stable training performance. Taking inspiration from the original WGAN paper (Arjovsky et al.), we also modified our model to train the discriminators at a higher frequency then the generator, which also led to improved quality in the output images.

## 6 Evaluation

In this section we describe our dataset, experimental setup, and results for MiddleGAN across multiple datasets.

### 6.1 Experimental Setup

**Dataset:** We primarily evaluated MiddleGAN using the CelebA dataset (Liu et al., 2015), which was proposed by Liu et al. in 2015. The CelebA dataset contains over 200,000 annotated images of celebrity faces. We selected this dataset due to its large size, extensive use in prior works, and separability into two domains ("Male" and "Female"). To create the two domains we used the boolean "Male" identifier present in the original dataset annotations[1]. By splitting on the "Male" identifier we created two smaller datasets, which we designated 'Male' (with approx. 84,000 images) and "Female" (with approx. 118,000 images). In all experiments, the "Female" dataset was assigned domain A, and the 'Male' dataset was assigned domain B. These dataset-domain assignments were based on alphabetical ordering, but given the order-agnostic structure of MiddleGAN with regards to input domains, we would anticipate identical results if the dataset-domain assignments were reversed.

**Image Generation:** To evaluate MiddleGAN, we used the "Male" and "Female" datasets to train nine different MiddleGAN models using a combination of three blend ratios (0.25, 0.50, and 0.75) and three image sizes (32px, 64px, and 128px). Aside from the blend ratio and Image Size, all models used the same hyperparameters as shown in Table 1. If our selection of a hyperparameter was heavily influenced by one

---

[1] All annotations in the CelebA dataset are booleans. As such, we made the assumption that any image that had a value of "1" for the "Male" identifier was male, and a value of "-1" was female. While this may not be 100% accurate, as a "-1" value is technically "Not Male" as opposed to "Female", we do not expect that this assumption significantly impacted our results.

Table 1: MiddleGAN Hyperparameters

| Hyperparameter | Description | Values | Source |
|---|---|---|---|
| Batch Size | Batch size during training. | 128 | DCGAN |
| NZ | Latent space for generator. | 128 | - |
| Epochs | Epochs during training. | 200 | - |
| $LR_D$ | Learning rate of discriminators. | 0.0001 | TransGAN |
| $LR_G$ | Learning rate of generator. | 0.0001 | TransGAN |
| $B_1$ | Beta 1 for Adam optimizers. | 0.0 | TransGAN |
| $B_2$ | Beta 2 for Adam optimizers. | 0.999 | DCGAN |
| N Critic | Discriminator to generator training ratio. | 5 | WGAN |
| GP Weight | Gradient penalty weight (used in loss function). | 100 | - |
| Blend Ratio | The blend ratio between input domains. | 0.25, 0.50, 0.75 | - |
| Image Size | The size of the generated images. | 32px, 64px, 128px | - |

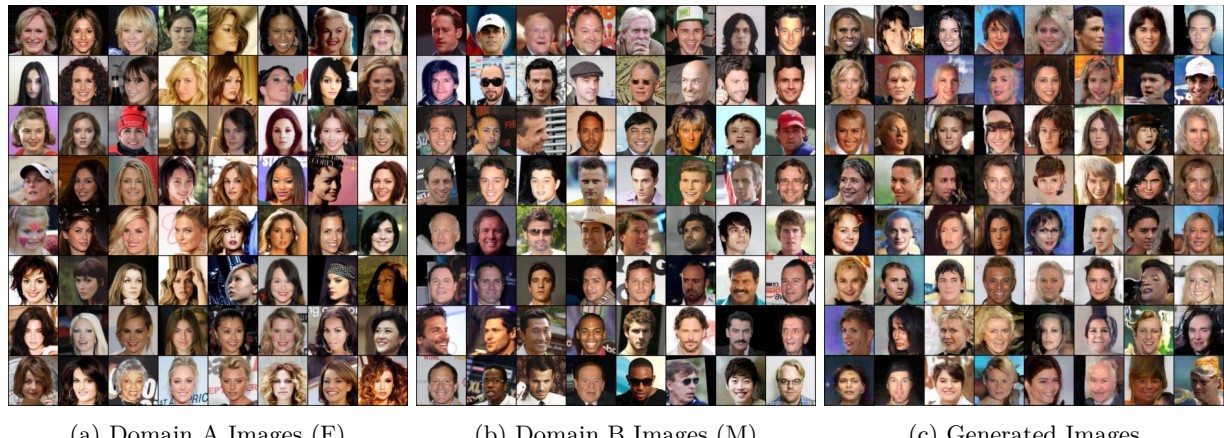

(a) Domain A Images (F)      (b) Domain B Images (M)      (c) Generated Images

Figure 4: This figure shows the results of images from both input domains as well as images generated by MiddleGAN. We used an image size of 128px and a blend ratio of 0.5. The generated images have both masculine and feminine qualities.

or more prior works, those works are noted in the "Source Paper" column. All models were trained on a NVIDIA A100 GPU, with an average per-model training time of around 12 hours. We generated 1000 images per model for use in our experiments.

**Feature Extraction & t-SNE:** We elected to use the pre-trained ResNet101 model provided by PyTorch as the basis of our feature extractor (pyt, c; He et al.). We then fine-tuned the model to provide a binary classification for male and female faces from the original CelebA dataset. This resulted in 3 fine-tuned ResNet101 models, one per Image Size. From there, we were able to extract feature vectors (length=2048) from the second-to-last layer of the ResNet101 model. We extracted feature vectors for each of the nine sets of images generated by MiddleGAN, as well as matching-size sets of images from both input domains. Roughly speaking, this means we extracted 3000 feature vectors for each of the nine models, which were evenly split between domain A images, domain B images, and generated images. Leveraging principal component analysis (PCA), we reduced the length of each extracted vector to 50 (skl, b). After performing PCA, we input the feature vectors into t-SNE in order to better understand the relationship between the input images and generated images.

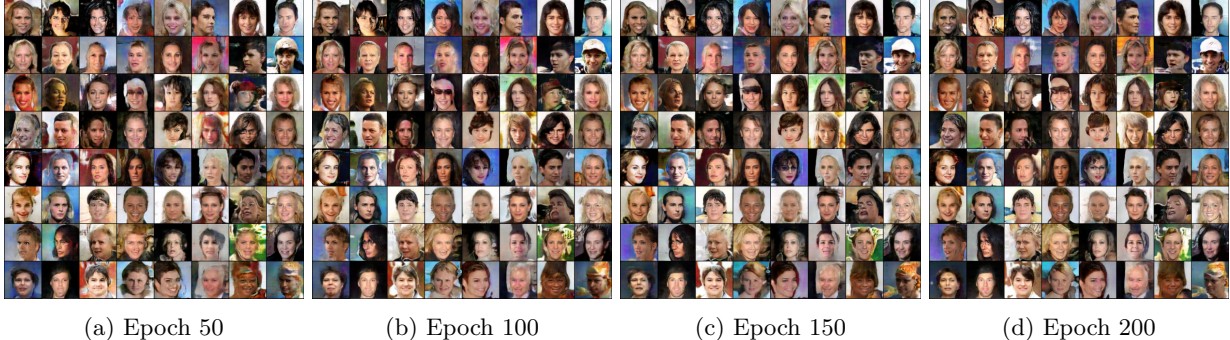

(a) Epoch 50       (b) Epoch 100       (c) Epoch 150       (d) Epoch 200

Figure 5: Generated images on fixed noise across 200 epochs. We used an image size of 128px and a blend ratio of 0.50. Image quality continues to improve until at least the 200th epoch.

## 6.2 Experimental Results

We achieved positive results on all nine trained versions of MiddleGAN, across a variety of blend ratios and image sizes. Figure 4 shows a sample of images from both input domains as well as images generated by MiddleGAN. An informal evaluation by the authors of the paper found the generated images to have both masculine and feminine qualities, which one would expect given the input domains of male and female faces. Beyond the mix of masculine and feminine qualities, we noted a diverse range of skintones and ethnicities present in both the input images and generated images, which further shows the diversity of images that MiddleGAN can generate. This is also an indicator that MiddleGAN did not suffer from mode collapse, which is when "the generator starts producing the same output (or a small set of outputs) over and over again" (pro).

While the training loss for the generator slowly increases over the course of the training, we continued to see improvements in the resulting imagery, as shown in Figure 5, which shows samples generated by MiddleGAN on fixed noise across training epochs. For this project, all training stopped after 200 epochs. We leave it to future work to evaluate if the quality of images generated by MiddleGAN could continue to improve after 200 epochs.

### 6.2.1 KID and FID Scores to Evaluate the Generated Images

The FID score is commonly used to measure how closely the distribution of generated images aligns with that of real images by comparing feature representations extracted from a pre-trained Inception network. It evaluates both the realism and diversity of the generated images, with lower FID values indicating a better match to real images. The KID score, in contrast, is an alternative metric that does not assume a Gaussian distribution for the features and is particularly beneficial when working with smaller datasets, as it avoids the bias that FID can introduce with limited sample sizes.

It's important to emphasize that FID and KID are typically applied in contexts where the goal is to generate images that closely mirror the real image distribution. However, our objective in this work is different. MiddleGAN is designed to generate blended inter-domain images that incorporate features from two distinct domains—male and female faces. As a result, the generated images are expected to fall somewhere between the two real distributions. Consequently, higher FID and KID values do not necessarily indicate poor performance but instead reflect the intentional design of our model. With this understanding, we present the computed FID and KID scores in Table 2.

Although the FID scores for the generated images compared to the male and female distributions are higher than what is typical for GANs designed to replicate a single domain, this is due to the specific nature of our task. Notably, FID(female, generated) < FID(female, male) and FID(male, generated) < FID(female, male), indicating that the generated images resemble both male and female faces more closely than male and female faces resemble each other. This demonstrates that MiddleGAN successfully captures the intermediate characteristics of both domains, producing the blended images as intended. The KID scores show a similar

|  | Female, Middle | Male, Middle | Female, Make |
|---|---|---|---|
| FID | 84.50 | 99.85 | 106.34 |
| KID | 0.0655 | 0.0700 | 0.0940 |

Table 2: The KID and FID scores to evaluate the generated images. It is worth highlighting that FID and KID are generally employed in tasks where the aim is to generate images that closely resemble the real image distribution. However, this differs from our objective in this work. MiddleGAN is specifically designed to produce blended inter-domain images that combine features from two distinct domains (male and female faces). As a result, the generated images are expected to fall between these two real distributions. Therefore, higher FID and KID scores do not necessarily signify poor performance but rather reflect the intentional design of our model.

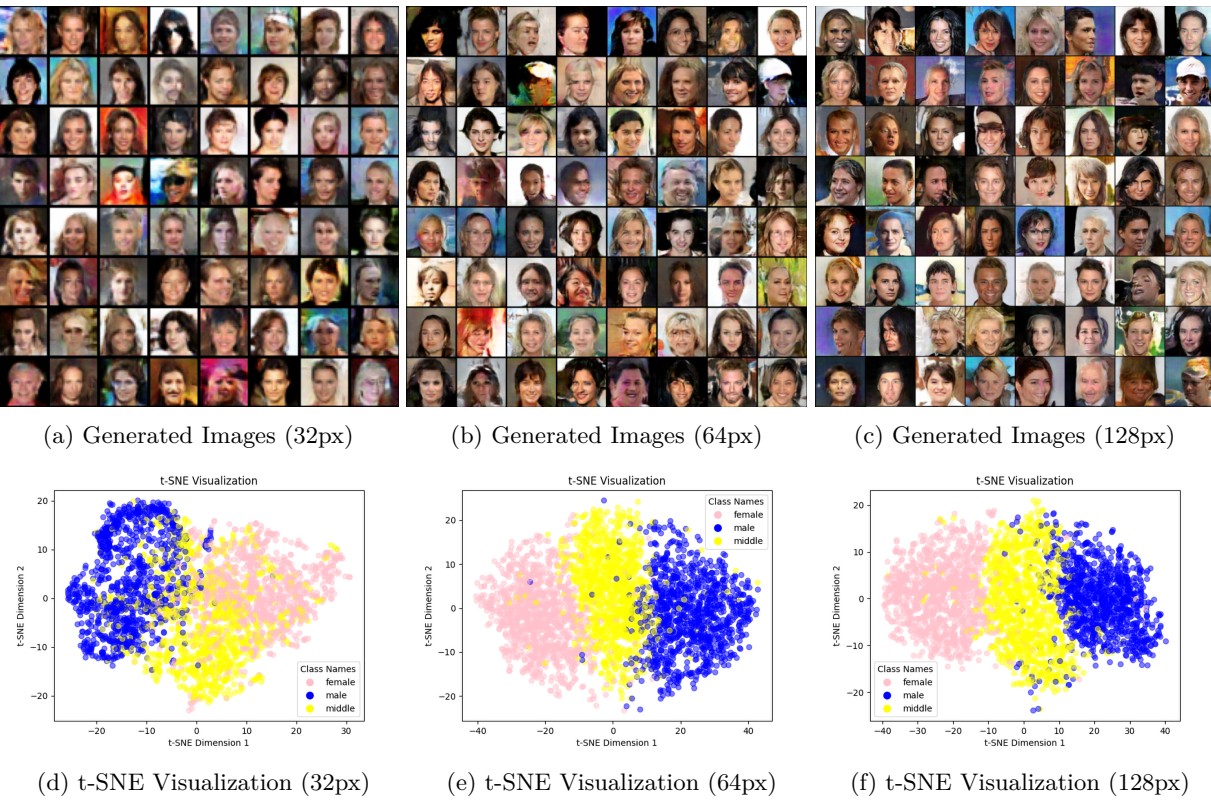

(a) Generated Images (32px)      (b) Generated Images (64px)      (c) Generated Images (128px)

(d) t-SNE Visualization (32px)   (e) t-SNE Visualization (64px)   (f) t-SNE Visualization (128px)

Figure 6: The impact of different image sizes on the images generated by MiddleGAN, with a blend ratio of 0.5. While the generated images contain masculine and feminine qualities at all resolutions, the t-SNE visualizations are noticeably different at lower resolutions. This is expected, as at low resolutions, there are not enough pixels to clearly differentiate between masculine and feminine qualities.

trend, further confirming that the generated images capture the diversity and overlap of features between the two domains, rather than merely replicating features from one domain in isolation. In conclusion, while FID and KID are traditionally used to assess how well generated images replicate real ones, in our case, these metrics reflect the inter-domain nature of the generated images, which are purposefully designed to blend features from both male and female faces. We hope this explanation clarifies our approach and the rationale behind the FID and KID results.

## 6.3 Image Size Experiments

The impact of different blend ratios on the images generated by MiddleGAN, with an image size of 128px. With a blend ratio of 0.25, the generated images were more masculine. When visualized with t-SNE, a

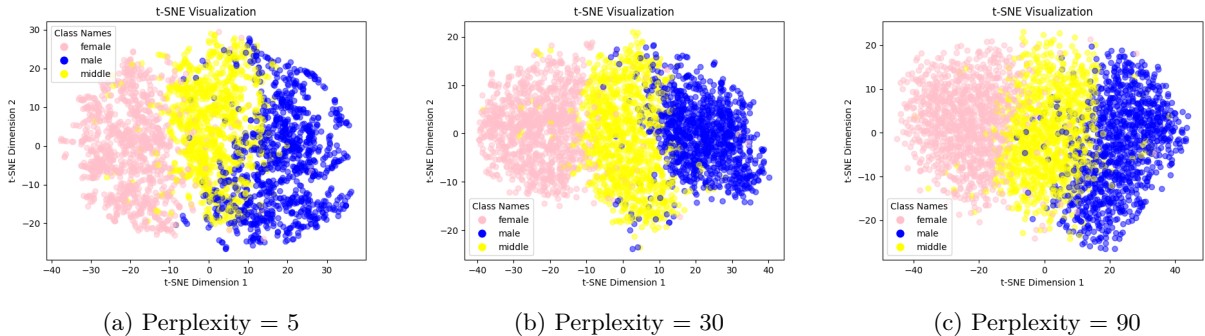

(a) Perplexity = 5          (b) Perplexity = 30          (c) Perplexity = 90

Figure 7: This figure shows the results of our t-SNE perplexity experiments. We used an image size of 128px and a blend ratio of 0.5. We varied the perplexity from 5 to 90. In all cases, the generated images fell within or between the input domain images, as expected.

blend-ratio of 0.25 caused the generated images to largely overlap with the images from the Male domain. A similar yet reversed pattern was visible with a blend ratio of 0.75, which resulted in more feminine images.

As previously discussed in Section 5, we modified the traditional DCGAN architecture to support any power-of-two image size by scaling the number of layers in the generator and discriminators. To assess the success of this modification, we evaluated MiddleGAN on three image sizes - 32px, 64px, and 128px. Figure 6 shows our results. Our results show that MiddleGAN can produce high quality images across a wide range of image sizes. Additionally, despite having different image resolutions (and thus layers), all three models had similar training losses and all had strong alignment when visualized with t-SNE. One key observation is that the generated samples fell *within* the distributions of the input domains at lower resolutions, while the generated samples fell *between* the distributions of the input domains at higher resolutions. We hypothesize that this is because at lower resolutions, there are not enough pixels to distinguish between masculine and feminine faces.

### 6.3.1 Blend Ratio Experiments

The inclusion of a blend ratio in our model architecture enabled greater flexibility and variety in the range of images that MiddleGAN could produce. In this section we explore the impact that the blend ratio had on generated images as well as the corresponding visualizations. We evaluated MiddleGAN on three blend ratios - 0.25, 0.50, and 0.75. Figure 1 shows our results along with samples of the original input images as reference.

When compared to a blend ratio of 0.50, the results shown for the other blend ratios are noticeably different. A blend ratio of 0.25 weighs the loss of discriminator A (which trains on Domain A, female images) at 25% and weighs the loss of discriminator B (which trains on Domain B, male images) at 75%. As such, the resulting images (shown in Figure 1a) are noticeably more male than images generated with a blend ratio of 0.5, while still being more feminine than the original all-male images. The training loss is also different, as discriminator B shows much lower losses than discriminator A. This intuitively makes sense, as the loss of discriminator B is more heavily penalized during training, and thus a priority for the model to minimize. Lastly, the t-SNE visualization shows a shift in distribution, with the generated images falling both within and between the male and female images, as shown in Figure 1d. A blend ratio of 0.75 shows similar but inverse changes, with the images, training losses, and t-SNE visualizations all being biased towards female images (shown in Figures 1c, and 1f).

### 6.4 t-SNE & Perplexity Results

Given our reliance on t-SNE as an indicator for the quality of the images generated by MiddleGAN, we wanted to ensure that any results that we observed were not due to chance in terms of hyperparameter selection. Of

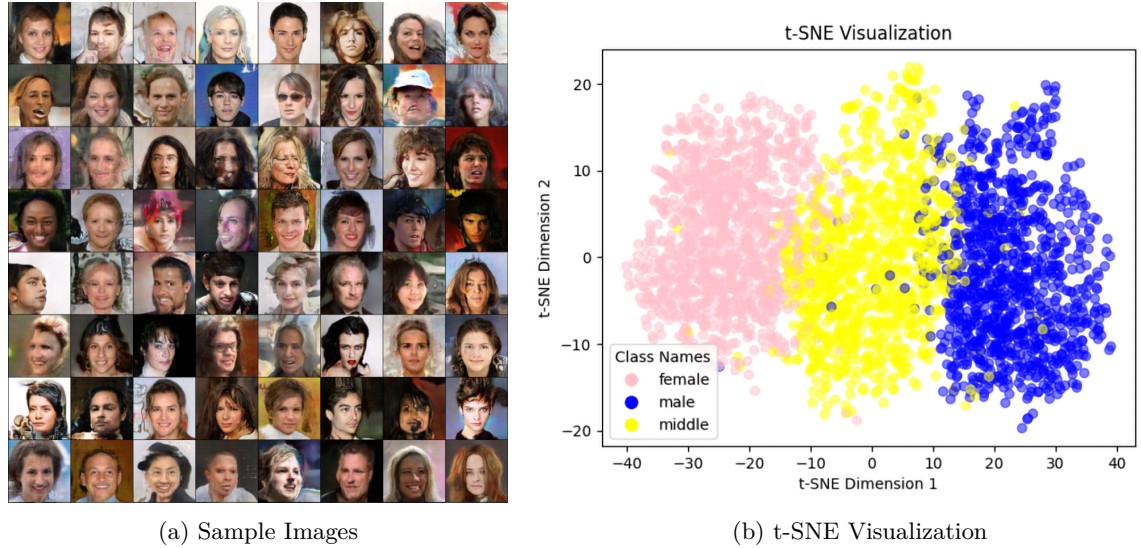

(a) Sample Images    (b) t-SNE Visualization

Figure 8: The preliminary results from our high resolution (256px) implementation of MiddleGAN. The generated images have both masculine and feminine qualities when using a blend ratio of 0.50. Additionally, the generated images fall within the input domains when visualized with t-SNE.

specific concern to us was the "Perplexity" hyperparameter, which past research has established can have a dramatic effect on the visualizations that t-SNE generates (Wattenberg et al., 2016; skl, a).

Figure 7 shows the results of three different perplexity experiments we performed, with perplexity values ranging from 5 to 90. We selected this range of values based on the guidance provided in the scikit-learn implementation of t-SNE, which recommends "a value between 5 and 50" (skl, c). For these experiments, all input images were 128px resolution images, with the generated images created with a blend ratio of 0.5. Our results show that perplexity did not have a major impact on the t-SNE visualizations we generated, which is a positive result. We ultimately selected a perplexity value of 30 for all of our other t-SNE visualizations presented in this paper, as we found it generated consistently good results. Coincidentally, the default Perplexity value in scikit-learn's t-SNE implementation is 30, which supports our selection (skl, c).

## 6.5 High Resolution Results

While our evaluation of MiddleGAN's ability to scale to different image sizes presented in Section 6.3 is thorough, we were eager to see if MiddleGAN could produce high quality images beyond 128px resolution. The push to go beyond 128px resolution images was prompted by our observation that the original WGAN-GP paper appeared to only test image generation up to 128px resolution (Gulrajani et al.). Given our model's requirement for power-of-two image sizes, the next possible image size for us to generate was 256px resolution. Our initial 256px resolution results are shown in Figure 8. We believe these results, while preliminary, show a positive outcome with both the generated samples and the t-SNE visualization appearing as expected, even with the higher resolution.

## 6.6 Use Case for the Generated Inter-Domain Images

We present additional experiments using these images as training data for image classification tasks to evaluate their potential utility. Specifically, we applied the generated inter-domain images to a gender classification task on the CelebA dataset.

In the first experiment (Round 1), we trained a basic CNN to classify two genders (male and female) using a training set that included inter-domain images. The model was then tested on a standard test set containing images of both genders, achieving an accuracy of 89.51%. In the second experiment (Round 2), we expanded

|  | Without Inter-Domain Images | With Inter-Domain Images |
|---|---|---|
| Accuracy | 89.51% | 94.58% |

Table 3: The classification results on the CelebA dataset's testing set consisting of male and female faces, with and without the inter-domain images. With the data augmentation achieved via having the inter-domain images, we observe an increase in the classification accuracy score of 5.07%. This shows that the inter-domain images, when incorporated into the data augmentation strategy, can increase the performance of the image classification model.

the training set by adding a third category—'unsure'—to represent the inter-domain images that blend male and female features. The model was tested on the same two-gender test set from Round 1, but with an additional option: if the classifier identified an image as 'unsure,' it could opt not to classify it strictly as male or female. This approach led to an improved accuracy of 94.58%. These results indicate that incorporating synthetic inter-domain images can increase the flexibility and performance of image classification models by enabling them to better handle ambiguous cases, as observed in Table 3.

### 6.7 Handwritten Digits Experiments

Our evaluation of MiddleGAN for handwritten digit datasets followed a very similar process to that of the CelebA dataset. We first identified our datasets, trained MiddleGAN to generate images, and then visualized the results using t-SNE. We discuss our process and results below.

**Datasets:** While our earlier experiments split a single dataset into two domains, our handwritten digit experiments leverage two different datasets - each one serving as input ton one domain. We selected the MNIST dataset (LeCun et al., 2010), published in 2010, and the DIDA dataset (Kusetogullari et al., 2020b;a), published in 2020. The MNIST dataset contains over 70,000 black and white cropped and centered images of the digits 0 through 9. The DIDA dataset contains over 250,000 color images of the digits 0 through 9, sourced from Swedish historical documents from 1800 to 1940 - DIDA is claimed to be the "largest historical handwritten digit dataset" (Kusetogullari et al., 2020b;a). Unlike the CelebA experiments, in which the domains were both relatively similar (human faces), the DIDA and MNIST datasets are visually distinct and have different levels of complexity (the images in MNIST are at least 3 times less complex then DIDA due to having a single channel of black and white color information).

**Image Generation:** We trained MiddleGAN using the DIDA and MNIST datasets, specifically images from the class of "7" for both datasets. Samples of both input datasets are shown in Figure 9. We utilized a image size of 32px and a blend ratio of 0.60. Given that handwritten digits are significantly simpler then human faces, we modified many of the hyperparameters shown in Table 1 to better work for handwritten digits. We selected a blend ratio of 0.60 after observing that a blend ratio of 0.50 caused MiddleGAN to only produce MNIST-like images. By adjusting the blend ratio to 0.60 (with DIDA as domain A), we placed more weight on the more complex dataset, which ensured a diverse quality of images were produced when MiddleGAN was trained. This observation is important, as it means that the blend ratio can be used to compensate for cases when two domains of dataset have radically different complexities.

**Feature Extraction & t-SNE:** We elected to use the pre-trained GoogleLeNet model provided by PyTorch as the basis of our feature extractor (pyt, b; Szegedy et al., 2014). We opted for GoogleLeNet over ResNet101 as we felt GoogleLeNet was better suited for the simpler task of handwritten digit recognition. We fine-tuned the model to provide a binary classification for DIDA vs MNIST digits. From there, we were able to extract feature vectors from the second-to-last layer of the GoogleLeNet model. We extracted feature vectors for the images generated by MiddleGAN, as well as both input domains. After performing PCA, we input the feature vectors into t-SNE in order to better understand the relationship between the input images and generated images.

**Results:**

THe images generated by MiddleGAN during our handwritten digit experiments, as well as associated t-SNE visualization, are shown in Figure 10. While it is difficult to say if the generated images are truly in

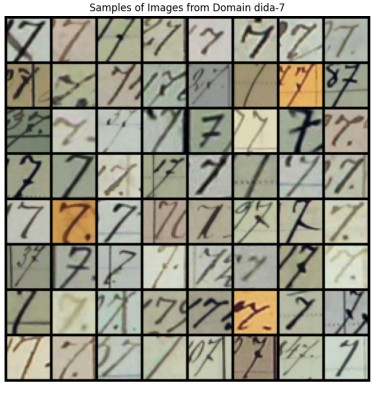

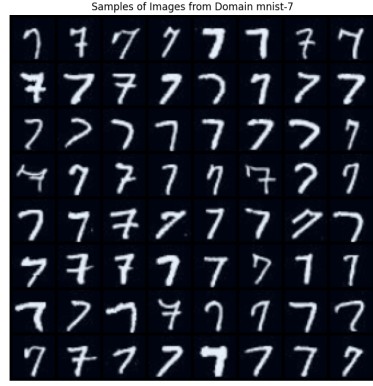

(a) Domain A Images (DIDA)    (b) Domain B Images (MNIST)

Figure 9: This figure shows images from both DIDA and MNIST input domains.

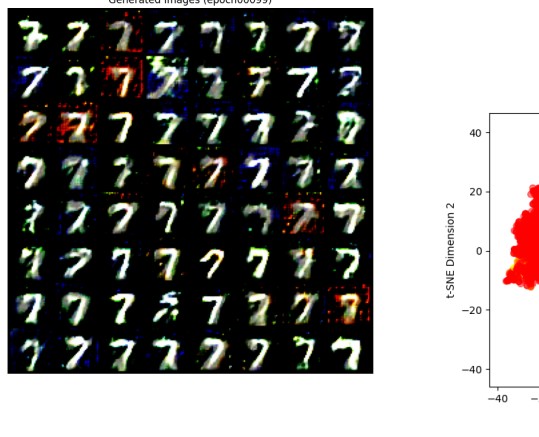

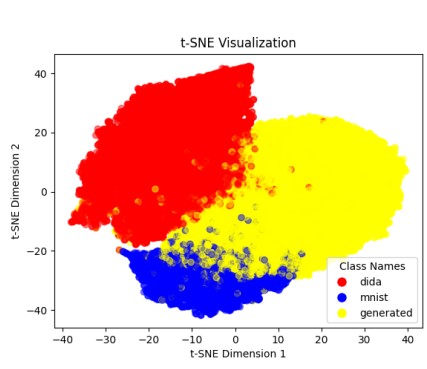

(a) Generated Images    (b) t-SNE Visualization

Figure 10: This figure shows the generated images for the blended DIDA-MNIST dataset as well as the associated t-SNE visualization.

the "middle" visually from a human perspective, the t-SNE visualization reveals that algorithmically the generated images fall between the DIDA and MNIST datasets. The results of this experiment emphasis the ability of MiddleGAN to blend sharply different domains together in a way that will still result in "middle" images when visualized with t-SNE.

# 7    Discussion & Future Work

In this paper we demonstrated both the theoretical and empirical results of images generated with Middle-GAN. However, we think there are many opportunities to build upon MiddleGAN. We discuss our existing work and opportunities for future work below.

### 7.1 MiddleGAN vs. Interpolating Label Embedding for Blending Generation.

One might be prompted to ask the question: Even though the proposed method is feasible, it is possible to simply train a GAN with two domains and interpolate the label embedding for blending generation. What would be the advantage of MiddleGAN? We explain that the advantages of MiddleGAN are:

- MiddleGAN explicitly trains its generator to produce images that blend features from two input domains by employing two separate discriminators, ensuring that the generated image is adversarially validated by both domain-specific discriminators. In contrast, label interpolation does not enforce inter-domain blending through adversarial training; instead, the generator is tasked with interpolating the learned embeddings, which may not guarantee a successful (adversarially validated) blending, as there is no explicit supervision over the blend produced by the interpolation.

- MiddleGAN's inter-domain images are designed to lie between the two distributions in the feature space, as validated by t-SNE visualizations and supported by theoretical results. The architecture is specifically structured to ensure that the generated images fall within this intermediate space, facilitating a smooth transition between the domains. Label interpolation, on the other hand, may not guarantee that the generated images occupy this in-between space in a controlled manner, as interpolation is often heuristic and not explicitly tied to the true data distributions.

- While label interpolation is an appealing approach and we agree that it is possible to train a GAN with two domains and interpolate the label embeddings to generate blended images, this method faces a potential issue known as catastrophic forgetting Kirkpatrick et al. (2017). This occurs when the model, trained sequentially on two domains, tends to forget the features learned from the first domain as it adapts to the second. This happens because the weights learned during training on the first domain can be overwritten during subsequent training on the second domain. MiddleGAN, in contrast, mitigates this problem by updating the weights of both discriminators and the generator simultaneously during training. This concurrent update ensures that both domains are continuously considered, preventing the overwriting of domain-specific features and thus avoiding catastrophic forgetting. As a result, MiddleGAN is able to generate blended images that more reliably and cohesively capture features from both input domains.

### 7.2 Comparision with Other Variations of the GAN as Baselines.

Traditionally, it is considered important to evaluate the performance of a new GAN variation by comparing it with other baseline models (existing variations of the GAN). However, we argue that traditional GAN models like DCGAN or StarGAN are not suitable for direct comparison in this case. These models are designed to generate images within a single domain or perform domain-to-domain translation, while MiddleGAN is specifically focused on generating inter-domain images that blend features from two distinct input sets.

Since models like DCGAN and StarGAN are not designed to produce inter-domain images, comparing them with MiddleGAN would not offer a meaningful assessment of the core contribution of our method—the ability to create blended images from distinct domains. Therefore, our experiments emphasize demonstrating the unique strengths of MiddleGAN in generating these images and applying them to tasks like classification, which traditional GANs are not equipped to address in the same way.

### 7.3 Instability on Training.

GANs are indeed known for their instability during training. To mitigate this, we implemented several stabilizing techniques, including the WGAN-GP loss function and adjustments to the training dynamics, such as increasing the discriminator-to-generator training ratio. These measures have resulted in noticeable improvements in training stability in our experiments. However, we acknowledge that further investigation into the stability of MiddleGAN is important. While our current approach has demonstrated stable training across various tasks, including inter-domain image generation, a more thorough exploration of how the model behaves under different hyperparameter settings or with additional discriminators would yield valuable insights.

In our experiments, we also observed a reduction in mode collapse by incorporating an adaptive weighted discriminator Zadorozhnyy et al. (2021), which contributed to more consistent convergence across multiple training runs. Initially, without the WGAN-GP loss function, mode collapse occurred frequently during training. However, after introducing both the WGAN-GP loss and the adaptive weighted discriminator, mode collapse was significantly reduced, with very few occurrences. These findings show that while progress has been made, there is still potential for further optimization to enhance stability.

### 7.4 Limitation of CelebA Dataset.

Throughout this work, we have primarily used the CelebA dataset with a split on the "male" attribute to create the "male" and "female" input domains. Although we achieved strong results using these domains, there are some inherent limitations of our approach and we caution against similar blending being blindly applied to future applications. Firstly, the CelebA dataset comprises images of celebrities, meaning our MiddleGAN model learns to generate faces resembling celebrities rather than everyday people. While we expect that MiddleGAN would perform equally well with photos of non-celebrities, it currently generates new celebrity-like faces. Secondly, and perhaps more importantly, MiddleGAN is learning stereotypical masculine and feminine traits rather than the true essence of being "male" or "female". This could potentially reinforce gender stereotypes if misused in the future, despite the model's ability to demonstrate the blending capabilities of MiddleGAN. We emphasize that MiddleGAN, like any tool, has the potential for artistic and other applications. However, it is crucial to remain mindful that certain blending combinations (e.g., different races) may produce uncomfortable or inappropriate results.

### 7.5 Expanding to N-Domains.

While MiddleGAN currently supports the blending of two-domains, this is not a hard limitation. In fact, MiddleGAN could be modified to support N-domains by modifying the training code to support one discriminator per input domain. However, while this is theoretically possible, we anticipate larger losses for, and the eventual collapse of, MiddeleGAN's generator if too many different domains were included. We anticipate this happening as the generator would be unable to create a single image that would satisfy all discriminators. Despite this anticipated challenge, we still believe that this area of research would be worth future exploration.

## 8 Conclusion

In this work, we introduced MiddleGAN, a novel variation of the traditional GAN capable of generating images in between two distinct input domains. MiddleGAN leverages two discriminators and one generator to create images that appear to be in both domains, resulting in blended images. The theoretical basis for MiddleGAN was covered in detail, in additional to the empirical evaluation. We extensively evaluated the capabilities of MiddleGAN on the CelebA dataset across three Blend Ratios (0.25, 0.50, and 0.75) and three Image Sizes (32px, 64px, and 128px). Our evaluation with the CelebA dataset showed the blended nature of the generated images, which contained both masculine and feminine features. In our handwritten digit based experiments, we revealed that with some adjustments of the Blend Ratio hyperparameter, MiddleGAN could handle datasets with radically different complexities (MNIST being "low" complexity and DIDA being "high" complexity). We concluded our work by outlining future opportunities for MiddleGAN as well as the limitations and caveats of MiddleGAN.

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
