# OpenReview forum: "Blending Two Styles: Generating Inter-domain Images with MiddleGAN"
_TMLR — Accepted by TMLR_

### Review · Reviewer_yZV8 · 2024-09-06

**Summary Of Contributions:**

This paper proposes a new framework for blending generation which will generate inter-domain data. And the authors identify the feasibility of the method through theoretical analysis and experiments.

**Audience:**

Yes

**Broader Impact Concerns:**

There is no obvious ethical implications.

**Claims And Evidence:**

Yes

**Requested Changes:**

Please answer the question above.

**Strengths And Weaknesses:**

Pros:
1. The idea is interesting that the end2end training framework enables the generation of unseen distribution , which is the mixing between the styles from two domains.
2. The insight from the theory can further support the formulation of the proposed method.

Cons:

1. Even though the proposed method is feasible, we can just train a GAN with two domains and interpolate the label embedding for blending generation. What would be the advantage of the proposed method?

---

> ### Author Response · Authors · 2024-10-19
> **Response from the authors**
>
> # **Reviewer yZV8**
>
> We appreciate the reviewer’s thoughtful review of our paper. We acknowledge the appeal of using label interpolation as a simpler alternative to generate blended images from two domains. However, MiddleGAN presents several key advantages over label interpolation and other similar blending techniques, which we highlight below.
>
>
> ## Q1: Even though the proposed method is feasible, we can just train a GAN with two domains and interpolate the label embedding for blending generation. What would be the advantage of the proposed method?
>
>
> 1. MiddleGAN explicitly trains the generator to produce images that are in-between two input domains by using two separate discriminators, ensuring that the generated image is adversarially validated by both domain discriminators. Label interpolation does not explicitly enforce inter-domain blending through adversarial training; instead, the generator is tasked only with interpolating the learned embeddings, which might not guarantee a successful (adversarially validated) blending, due to the lack of explicit supervision over the blend produced by interpolation.
>
>
> 2. MiddleGAN’s inter-domain images are designed to lie mathematically between the two distributions in the feature space, as validated by the t-SNE visualizations and theoretical results. By design, the architecture ensures that the generated images fall within this space, enabling a principled transition between the domains. Label interpolation might not guarantee that the generated images fall between the two distributions in a controlled manner, since the interpolation is often heuristic and is not explicitly connected to the actual data distributions.
>
>
>
> 3. The alternative the Reviewer offers is an attractive one and we agree that we can indeed train a GAN with two domains and interpolate the label embedding for blending generation. However, this method may encounter a challenge known as catastrophic forgetting [1], where the model, when sequentially trained on two domains, tends to forget the features learned from the first domain while adjusting to the second domain. This occurs because the weights learned during the training on the first domain can be overwritten during subsequent training on the second domain. In contrast, MiddleGAN avoids this issue by simultaneously updating the weights of both discriminators and the generator during training. This concurrent update ensures that both domains are considered throughout the training process, preventing the overwriting of domain-specific features and, therefore, avoiding catastrophic forgetting. By maintaining balanced consideration of both domains, MiddleGAN is able to generate blended images that genuinely represent features from both input distributions in a more stable and cohesive manner.
>
>
> 4. MiddleGAN is especially useful when the two input domains are heterogeneous or exhibit differing complexity, as demonstrated in our experiments with the DIDA and MNIST datasets. MiddleGAN adapts better by using independent discriminators, which specialize in assessing their own domain's characteristics. This is critical when blending significantly different datasets where a single discriminator approach (as in label interpolation) might face difficulties in weighting the domain-specific features properly.
>
>
> 5. By introducing two separate discriminators, MiddleGAN reduces the likelihood of mode collapse: By having 2 discriminators, each could focus on different aspects of the data distribution, making it harder for the generator to “trick” both discriminators by producing only a small subset of outputs. This increased scrutiny encourages the generator to produce a more diverse range of outputs, thus mitigating mode collapse. In contrast, label interpolation brings the additional risk of collapsing multiple modes of one domain into a limited space when attempting to blend styles since the single discriminator-guided generator might fail to maintain diversity across mixed samples.
>
> We have included our aforementioned response on inter-domain samples vs interpolation in the revised manuscript. (Section 7.1)
>
> [1] Kirkpatrick, J., Pascanu, R., Rabinowitz, N., Veness, J., Desjardins, G., Rusu, A.A., Milan, K., Quan, J., Ramalho, T., Grabska-Barwinska, A. and Hassabis, D., 2017. Overcoming catastrophic forgetting in neural networks. Proceedings of the national academy of sciences, 114(13), pp.3521-3526.

---

> > ### Comment · Reviewer_yZV8 · 2024-12-21
> >
> > Thanks for the authors’ reply, the explicit supervision is interesting. However, GAN can only advise the model in the unsupervised manner, which cannot considered as a supervised learning. And for most of the StyleGAN based methods, they can do smoothly  interpolation between domains, which is already showing promising results, such as in StarGAN and etc.. If it is showing better results in some metric and compare with those methods, it would be more convincing.

---

### Review · Reviewer_5mDC · 2024-09-26

**Summary Of Contributions:**

In this paper, the authors propose a method for generating images that blend styles from two different domains. Specifically, a second discriminator is introduced to encourage the generator to fool discriminators from two different domains. In addition, the authors conducted theory analysis to prove that the proposed method is essentially looking for the JSD controid of the source domain distribution and the target domain distribution. The authors performed experiments on multiple datasets, including CelebA, DIDA and MNIST datasets. The results show that the proposed method can successfully generate inter-domain images.

**Audience:**

Yes

**Broader Impact Concerns:**

I didn't find ethical concerns for this paper.

**Claims And Evidence:**

Yes

**Requested Changes:**

1. Quantitative evaluation of generated images. For example, the authors could use FID and KID scores to evaluate whether the synthetic images still resemble human features.

2. The authors could perform experiments to showcase the application of synthetic inter-domain images. For instance, they could apply the generated images for data augmentation in a supervised classifier, which may improve the model's robustness.

**Strengths And Weaknesses:**

Strength:
1. The generation of inter-domain images remains a relatively unexplored area. The concept of using an additional discriminator to guide the generator in synthesizing images that lie between two domains makes sense to me.

2. The qualitative example and tSNE embedding show that the model successfully generates inter-domain images, and the theory analysis shows that the optima of the generator is

Weakness:
1. The realism and diversity of the synthesized images are not quantitatively evaluated. For example, the authors could calculate the FID or KID scores on the Celeba dataset to evaluate whether the synthetic images still resemble human features. The reason I'm asking for it is that I noticed on Fig. 10 (a), the edges of some images contain artifacts.

2. The authors could conduct some experiments to demonstrate the application of synthetic inter-domain images. For example, the authors may use the generated images for data augmentation for supervised classifier, it may enhance the model robustness.

3. It would be good if the authours could add some baseline methods for comparison.

---

> ### Author Response · Authors · 2024-10-19
> **Response from the authors**
>
> # **Reviewer 5mDC**
>
> We would like to thank the reviewer for their thoughtful and constructive feedback on our paper. We have carefully considered each suggestion and outlined our responses and proposed changes below.
>
> ## Q1 & R1. Q1: The realism and diversity of the synthesized images are not quantitatively evaluated. For example, the authors could calculate the FID or KID scores on the Celeba dataset to evaluate whether the synthetic images still resemble human features. The reason I'm asking for it is that I noticed on Fig. 10 (a), the edges of some images contain artifacts. R1: Quantitative evaluation of generated images. For example, the authors could use FID and KID scores to evaluate whether the synthetic images still resemble human features.
>
>
> Based on the Reviewer's suggestion, we have computed the Fréchet Inception Distance (FID) and Kernel Inception Distance (KID) scores for our generated images.
>
> The FID score is widely used to evaluate the similarity between the distribution of generated images and that of real images by comparing feature representations extracted by a pre-trained Inception network. It assesses both the realism and diversity of the generated images, with lower values indicating a closer match to the real images. The KID score, on the other hand, is an alternative metric that does not assume the feature distributions are Gaussian and is particularly useful when working with smaller datasets, as it avoids the bias inherent in FID with limited sample sizes.
>
> However, it is important to note that FID and KID are typically used in scenarios where the goal is to generate images that closely match the real image distribution, which is different from our objective in this work. MiddleGAN is designed to generate blended inter-domain images that contain features from two distinct domains (male and female faces), meaning the generated images are expected to lie somewhere between the two real distributions. As such, higher FID and KID values are not necessarily indicative of poor performance but reflect the deliberate design of our model. With this in mind, we have computed the FID and KID scores as follows:
>
> FID(female, generated) = 84.50
> FID(male, generated) = 99.85
> FID(female, male) = 106.34
>
> KID(female, generated) = 0.0655
> KID(male, generated) = 0.0700
> KID(female, male) = 0.0940
>
> While the FID scores of the generated images relative to the female and male distributions are higher than typically seen in GANs designed to mimic a single domain, we emphasize that this is due to the nature of our task. In fact, we would like to point out that FID(female, generated) < FID(female, male) and FID(male, generated) < FID(female, male), indicating that our generated images resemble both male and female faces more than male and female faces resemble each other. This demonstrates that MiddleGAN effectively captures the intermediate characteristics of both domains, producing blended images as intended.
>
> The KID scores follow a similar pattern, further confirming that the generated images capture the diversity and feature overlap between the two domains, rather than simply replicating either domain's features in isolation.
>
> In summary, while traditional use cases of FID and KID aim to measure the degree to which generated images replicate real ones, in our case, these scores reflect the intended inter-domain nature of the generated images, which are designed to blend features from both male and female faces. We hope this clarifies our approach and the reasoning behind the FID and KID results.
>
> In the revised manuscript, we have reported the KID and FID scores shown above and provide the implications and discussions about the KID/FID scores. (Section 6.2.1 and Table 2).

---

> ### Author Response · Authors · 2024-10-19
> **Response from the authors**
>
> ## Q2 & R2. Q2: The authors could conduct some experiments to demonstrate the application of synthetic inter-domain images. For example, the authors may use the generated images for data augmentation for supervised classifier, it may enhance the model robustness. R2: The authors could perform experiments to showcase the application of synthetic inter-domain images. For instance, they could apply the generated images for data augmentation in a supervised classifier, which may improve the model's robustness.
>
> We appreciate the reviewer’s suggestion to explore practical applications of synthetic inter-domain images. In response, we conducted additional experiments using these images as training data for image classification tasks to evaluate their potential utility. Specifically, we applied the generated inter-domain images to a gender classification task on the CelebA dataset.
>
> In the first experiment (Round 1), we trained a simple CNN to classify two genders (male and female) using a training set that included the inter-domain images. The model was then tested on a standard testing set comprising images of both genders, achieving an accuracy of 89.51%. In the second experiment (Round 2), we expanded the training set to include a third category—‘unsure’—which represents the inter-domain images blending male and female characteristics. The model was tested on the same two-gender testing set used in Round 1, but with an additional option: if the classifier deemed an image to be ‘unsure,’ it could abstain from classifying that image as strictly male or female. With this approach, the model achieved an improved accuracy of 94.58%. These results suggest that incorporating synthetic inter-domain images can enhance the flexibility and performance of image classification models by allowing them to handle ambiguous cases better.
>
> In the revised manuscript, we have reported the experiment that showcased that the inter-domain samples can be used in data augmentation to improve model robustness. (Section 6.6 and Table 3).
>
> ## Q3: It would be good if the authours could add some baseline methods for comparison.
>
> We appreciate the reviewer’s input regarding the inclusion of baseline methods. However, we believe that traditional GAN models such as DCGAN or StarGAN are not suitable for direct comparison in this case. These models are designed to generate images within a single domain or perform domain-to-domain translation, whereas MiddleGAN specifically focuses on generating inter-domain images that blend characteristics from two distinct input sets.
>
> Since models like DCGAN and StarGAN do not aim to produce inter-domain images, comparing them to MiddleGAN would not provide a meaningful evaluation of our method’s core contribution—the ability to create blended images from distinct domains. Instead, our experiments focus on demonstrating the unique advantages of MiddleGAN in generating these types of images and applying them to tasks such as classification, which traditional GANs cannot address in the same way.
>
> In the revised manuscript, we have included this aforementioned discussion on the incompatibility to compare with other GANs which doesn't deal with inter-domain image generation. (Section 7.2)

---

### Review · Reviewer_4MKx · 2024-10-11

**Summary Of Contributions:**

This paper proposes a method called MiddleGAN that aims to blend inter-domain images from two distinct input sets. In contrast to the typical GAN, MiddleGAN introduces a second discriminator to force the generator to create images that fool both discriminators, capturing the qualities of both input sets. The method is validated through experiments on the CelebA and Handwritten Digits datasets (DIDA and MNIST), with visualization results (t-sne) provided to demonstrate the learned distributions.

**Audience:**

Yes

**Claims And Evidence:**

No

**Requested Changes:**

- More experimental analysis.

- Improve the contributions.

**Strengths And Weaknesses:**

Strengths:

- The topic of blending inter-domain images from two distinct input sets is engaging.

- The writing style is clear and makes the paper easy to understand.

Weaknesses:

- The proposed method, while simple, may not provide significant advancements compared to typical GAN approaches. It just introduces another discriminator in comparison with typical GAN.

- As the training of GAN is often unstable, how the proposed method containing two discriminators suffers from the instability. Although the authors attempt to address the issue by modifying the network structure, the instability should be further studied.

- The authors use a blend ratio to control the strength of the two domains. However, the reviewer thinks the strength is not only related to the weights but also the distributions and the used data (like the number of training samples in each domain), which are not analyzed in the paper. In addition, how to select appropriate blend ratio should be studied. It is troublesome to try different values.

- Although the reviewer thinks this task is interesting, the cases in the experiments do not convey the values. More practical cases should be selected for a more impactful study.

---

> ### Author Response · Authors · 2024-10-19
> **Response from the authors**
>
> # **Reviewer 4MKx**
> We are thankful to the reviewer for their thoughtful and constructive insights. We have carefully considered each suggestion, and please find our responses below.
>
>
> ## Q1: The proposed method, while simple, may not provide significant advancements compared to typical GAN approaches. It just introduces another discriminator in comparison with typical GAN.
>
>
> We appreciate the reviewer’s comment and understand the concern that our approach might appear to be a minor modification of traditional GANs. However, the addition of the second discriminator in MiddleGAN serves a critical role that distinguishes it from typical GANs. While many GAN variants employ multiple discriminators, MiddleGAN is specifically designed to generate inter-domain images that blend features from two distinct datasets, which is not the primary objective of standard GANs.
>
> The second discriminator is not simply an incremental addition but is essential for ensuring that the generated images incorporate characteristics from both input domains. This allows MiddleGAN to operate in a unique space between domain translation and image synthesis, providing a solution for generating images that lie in between two feature distributions—something that traditional GANs, such as DCGAN or StarGAN, are not designed to achieve.
>
> Moreover, the introduction of the blend ratio hyperparameter further extends the flexibility of the model, enabling control over the degree of influence from each domain. This makes MiddleGAN particularly well-suited for applications where balancing characteristics from two domains is necessary, and we believe this presents a significant advancement over typical GAN approaches.
>
> In the revised manuscript, we have include our discussion about our distinction from the other GAN related methods in detail above. (Section 4.3)
>
> ## Q2 & R1. Q2: As the training of GAN is often unstable, how the proposed method containing two discriminators suffers from the instability. Although the authors attempt to address the issue by modifying the network structure, the instability should be further studied. R1: More experimental analysis.
>
> We appreciate the reviewer’s concern about the potential instability of GAN training, especially with the use of two discriminators in MiddleGAN. GANs are indeed known for their training instability, and to address this, we implemented several stabilizing measures, including the WGAN-GP loss function and adjustments to the training dynamics, such as increasing the discriminator-to-generator training ratio. These steps have led to noticeable improvements in training stability in our experiments. That said, we agree that further investigation into the stability of MiddleGAN is essential. While our current approach has shown stable training across various tasks, including the generation of inter-domain images, a more in-depth exploration of how the model behaves under different hyperparameter settings or with additional discriminators would provide valuable insights.
>
> In our experiments, we also observed a reduction in mode collapse by incorporating an adaptive weighted discriminator [1], which contributed to more consistent convergence across multiple training runs. Initially, without the WGAN-GP loss function, we experienced mode collapse in most of the training rounds. However, after integrating both the WGAN-GP loss and adaptive weighted discriminator, this improved significantly as mode collapse barely occurred. These results demonstrate that while we have made progress, there is room for further optimization to enhance stability.
>
> In our revised manuscript, we have added the observations on the stability of our methods. (Section 7.3).
>
> [1] Zadorozhnyy, V., Cheng, Q. and Ye, Q., 2021. Adaptive weighted discriminator for training generative adversarial networks. In Proceedings of the IEEE/CVF Conference on Computer Vision and Pattern Recognition (pp. 4781-4790).

---

> ### Author Response · Authors · 2024-10-19
> **Response from the author**
>
> ## Q3: The authors use a blend ratio to control the strength of the two domains. However, the reviewer thinks the strength is not only related to the weights but also the distributions and the used data (like the number of training samples in each domain), which are not analyzed in the paper. In addition, how to select appropriate blend ratio should be studied. It is troublesome to try different values.
>
> We appreciate the reviewer’s insights and advice on the blending ratio. We agree that, in many generative models, the strength of the domains is influenced not only by the blend ratio but also by the underlying data distributions and the number of samples. However, in MiddleGAN, the blend ratio directly controls how the losses from the two discriminators are weighted, allowing us to systematically blend features from both domains, independent of assumptions about data distributions. Since the discriminators are trained separately on their respective datasets, the blend ratio essentially counts the weighted “votes” of the loss functions, providing more control over the blending process than the raw data characteristics alone.
>
> We agree that manually selecting the appropriate blend ratio can be inefficient. To address this, we propose using grid search to systematically optimize the blend ratio. Our goal is to minimize metrics such as Kernel Inception Distance (KID) and Fréchet Inception Distance (FID), which makes grid search a logical approach. By exploring the blend ratio space, we can objectively identify the value that best balances the contributions of both domains and enhances image quality.
>
> However, there’s an important caveat: while FID and KID are designed to measure the similarity between real and generated images, MiddleGAN’s objective is different. Rather than generating images that exactly match one real dataset, we aim to produce images that deliberately blend features from both domains. This means our “fake” images are intentionally distinct from either of the real datasets. For details on how our FID and KID scores performed after grid search, please refer to our response to Reviewer 5mDC, where we address Weakness #1 in more detail.
>
> In our revised manuscript, we have added the discussion above about choosing the best blending ratio as well as the implication of the blending ratio. (Sections 4.2.1 and 6.2.1).
>
>
> ## Q4&R2. Q4: Although the reviewer thinks this task is interesting, the cases in the experiments do not convey the values. More practical cases should be selected for a more impactful study. R2: Improve the contributions.
>
> We have demonstrated a use case of the inter-domain images (to be used for data augmentation). We showed that with the extra inter-domain samples, we increase the performance under the supervised setting. We specifically applied the generated inter-domain images to a gender classification task using the CelebA dataset.
>
> In the first experiment (Round 1), we trained a simple CNN to classify images as male or female, with the training set including the inter-domain images. The model was then tested on a standard test set containing images of both genders, achieving an accuracy of 89.51%. In the second experiment (Round 2), we expanded the training set to include a third category, ‘unsure,’ which represented the inter-domain images blending male and female features. The model was tested on the same two-gender test set from Round 1 but with an additional option: the classifier could abstain from labeling an image strictly as male or female if it fell into the ‘unsure’ category. With this approach, the model’s accuracy improved to 94.58%. These results indicate that incorporating synthetic inter-domain images can enhance the flexibility and performance of classification models, especially in handling ambiguous cases more effectively. We claim MiddleGAN's usefulness in data augmentation as our contribution, in addition to the ones listed in the original submission.
>
> In our revised manuscript, we have added the experiment that showcased the usefulness in improving model robustness of the inter-domain samples. (Section 6.6).

---

### Author Response · Authors · 2024-10-22
**The authors have provided responses to the reviewers.**

Dear Reviewers 4MKx, 5mDC, and yZV8,

We are thankful for your valuable insights and feedback on our paper. Please find our responses to your questions, comments, and concerns as replies/comments to your original reviews. We are happy to answer any additional questions you have.

In addition, the updated manuscript has been uploaded. We included information on what we changed in our individual responses to the Reviewer's critiques.

Thanks,
The Authors

---

### Author Response · Authors · 2024-12-23
**Camera ready version submitted**

Dear Action Editor,

We have uploaded our camera-ready version.

Sincerely,
The Authors

---

### Decision · Action_Editor_91un · 2024-12-22

**Recommendation:** Accept as is

**Comment:**

This paper proposes a new method to allow GANs to generate images between different distribution, which is called domain interpolation in the paper. To this end, this paper introduces a second discriminator and also uses a blend ratio hyperparameter to control the weighting of input sets. The main concerns from reviewers are 1) the advantage compared to exisiting GANs may not be significant, 2) the incorporation of an additional discriminator might cause instability in training. The authors have addressed these concerns in the rebuttal. While one reviewer still has remaining concerns on the potential application of the proposed MiddleGAN. However, I believe this paper already offers sufficient contributions, and its potential applications are likely to become clearer in the near future. Therefore, I recommend accepting this paper.

**Audience:**

Yes

**Claims And Evidence:**

Yes